# Nanocomposite Film Based on Cellulose Acetate and Lignin-Rich Rice Straw Nanofibers

**DOI:** 10.3390/ma12040595

**Published:** 2019-02-16

**Authors:** Mohammad Hassan, Linn Berglund, Ragab Abou-Zeid, Enas Hassan, Wafaa Abou-Elseoud, Kristiina Oksman

**Affiliations:** 1Cellulose and Paper Department & Centre of Excellence for Advanced Sciences, National Research Centre, 33 El-Behouth street, Dokki, Giza 12622, Egypt; r_abouzeid2002@yahoo.com (R.A.-Z.); enas.elgarhy@nrc.sci.eg (E.H.); nrc_wafaa2014@yahoo.com (W.A.-E.); 2Egypt Nanotechnology Centre, Cairo University, El-Sheikh Zayed, 6th October City 12588, Egypt; 3Department of Engineering Sciences and Mathematics, Luleå University of Technology, SE 97187 Luleå, Sweden; linn.berglund@ltu.se; 4Fibre and Particle Engineering, University of Oulu, FI-90014 Oulu, Finland

**Keywords:** nanofibers, rice straw, unbleached pulp, cellulose acetate, nanocomposites

## Abstract

Nanofibers isolated from unbleached neutral sulfite rice straw pulp were used to prepare transparent films without the need to modify the isolated rice straw nanofibers (RSNF). RSNF with loading from 1.25 to 10 wt.% were mixed with cellulose acetate (CA) solution in acetone and films were formed by casting. The films were characterized regarding their transparency and light transmittance, microstructure, mechanical properties, crystallinity, water contact angle, porosity, water vapor permeability, and thermal properties. The results showed good dispersion of RSNF in CA matrix and films with good transparency and homogeneity could be prepared at RSNF loadings of less than 5%. As shown from contact angle and atomic force microscopy (AFM) measurements, the RSNF resulted in increased hydrophilic nature and roughness of the films. No significant improvement in tensile strength and Young’s modulus was recorded as a result of adding RSNF to CA. Addition of the RSNF did not significantly affect the porosity, crystallinity and melting temperature of CA, but slightly increased its glass transition temperature.

## 1. Introduction

Cellulose acetate (CA) is one of the earliest cellulose derivatives used in different applications including membranes for water treatment [1,2,3,4,5], gas separation [6,7], films for packaging [8,9,10], and fibers in textile industry [11]. CA has several attractive advantages including availability of cellulose from different resources, ease of processing, good mechanical properties, biodegradability, and high transparency.

Nanomaterials have been used to impart CA new characteristics such as antimicrobial properties [12], bioactivity [13], photoactivity [14,15,16], UV shielding [17], magnetic [18], catalytic activity [19], gas permeability [6], and flame retardance [20,21]. An important property of the nanomaterials used is their ability to keep transparency of CA films.

Nanocelluloses, e.g., cellulose nanofibers (CNF) and cellulose nanocrystals (CNC), are usually prepared from bleached cellulose pulps and are characterized by high mechanical properties, transparency in different polymer matrices, and hydrophilic properties. The blending of nanocelluloses with other polymers requires good compatibility between them, which is not always the case due to the relatively low hydrophilic nature of some polymers, such as CA, and the high hydrophilic nature of nanocelluloses. Therefore, chemical modification of nanocellulose surfaces with hydrophobic moieties has been studied to prepare CA/nanocellulose films with enhanced mechanical properties and transparency. Regarding modification of CNC for mixing with CA, modification with 3,3′,4,4′-benzophenone tetracarboxylic dianhydride has been recently reported [22]. Nevertheless, thanks to the ability to disperse CNC in some polar aprotic solvents [23], the use of non-modified CNC with CA has been reported [24,25,26,27,28,29]. Water of CNC aqueous suspension was first removed and exchanged by another polar aprotic solvent such as such as DMF, DMSO, 1,3-dimethyl-2-imidazolidinone (DMI), N,N-dimethylacetamide (DMAc), and 1-methyl-2-pyrrolidinone (NMP), then CNC was dispersed by applying ultrasonic or mechanical treatment. In the aforementioned examples, enhancement of mechanical properties and hydrophilicity of CA could be achieved by the addition of CNC.

On the other hand, the use of CNF isolated from bleached pulps with CA is somewhat different to using CNC. This is due to differences in the ease of dispersion in aprotic polar solvents because of the longer length and higher hydrophilicity of CNF. Accordingly, much fewer studies have been published so far on using CNF with CA without modification of the former. In addition, the previously prepared nanocomposite films were formed by phase inversion (not by direct casting) and consequently the films were white in color, opaque, highly porous and asymmetric. For example, CNF in water suspension was added to a CA acetone/water solution and films were formed by non-solvent phase inversion [30]; the films were tested as membranes regarding their water flux and ability to clarify fruit juice and whey. In another work, TEMPO-oxidized cellulose nanofibers (up to 2.5% based on CA) were mixed with CA after solvent exchange of the nanofibers with dimethyl formamide [31,32]. The films were formed by non-solvent phase inversion and evaluated as membranes regarding water flux and ultrafiltration characteristics. To prepare transparent CA/CNF film by casting (without phase inversion), CNF was chemically modified with 3-aminopropyltriethoxysilane before mixing with CA [33]. The Young’s modulus and tensile strength of CA films increased from 1.9 GPa to 4.1 GPa upon the addition of 7.5% of the modified CNF while tensile strength increased from 38 MPa to a maximum of 63.5 MPa at 2.5% CNF addition.

There is a recent trend in producing CNF from unbleached cellulose pulp to obtain nanofibers with a lignin-rich surface [34]. Investigating the properties of nanocomposites produced using nanofibers with a high lignin content is thus important to see what the advantages and disadvantages of these nanofibers are. To the best of our knowledge, the use of cellulose nanofibers with a high lignin content to make cellulose acetate nanocomposites films by casting without chemical modification of the nanofibers has not been studied so far. Using cellulose nanofibers isolated from unbleached pulp in composites reduces the cost when compared to those isolated from bleached pulps. In addition, the presence of lignin covering the surface of cellulose nanofibers makes their surface properties different from those isolated from bleached pulps, and thus affects the properties of nanocomposites. The presence of lignin at the surface of the nanofibers can also improve the dispersion of the cellulose nanofibers in polar aprotic solvents, and also improve compatibility with relatively less hydrophilic polymers, such as CA. In addition, the presence of lignin with nanofibers allows for easier removal of water and its exchange with polar aprotic solvents using simple washing rather than the exhaustive solvent exchange needed in case of nanofibers obtained from bleached pulps.

Therefore, the aim of the current work was to study the blending of CA with RSNF isolated from unbleached neutral sulfite pulp—i.e., nanofibers with a lignin-rich surface—to make films with high homogeneity and transparency. The effect of RSNF addition on mechanical, thermal, and crystallinity properties, and the water contact angle of CA films was studied.

## 2. Experimental

### 2.1. Materials

Rice straw obtained from a local farm in Qalubiyah, Egypt. Reagent grade sodium sulfite, sodium carbonate, citric acid, and sodium citrate were used as received from Sigma-Aldrich (Sigma-aldrich, St. Louis, MO, USA). Xylanase powder, ≥2500 units/g, recombinant, expressed in *Aspergillus oryzae* was purchased from Sigma-Aldrich and used as received. Cellulose acetate with ~39.8 wt.% acetyl content and average molecular weight M_n_ ~30,000 was used as received from Sigma-Aldrich.

### 2.2. Preparation of Rice Straw Pulp

Rice straw neutral sulfite pulp was prepared as previously described [34] using 10% sodium sulfite and 2% sodium carbonate (based on weight of rice straw) solutions at 160 °C for 2 h; the liquor ratio was 1:10. The pulp was washed with water, defibrillated in a Valley beater (Valley Iron Works, Appleton, WI, USA) to a 25° Schopper-Riegler (SR) degree of freeness, dewatered, and allowed to air dry. The chemical composition of the pulp was analyzed according to the standard methods [35] and was 54.1% α-cellulose, 14.2% Klason lignin, 3.2% acid insoluble lignin, 14.3% pentosans, and 16.6% ash content. The degree of polymerization (DP) was measured using cupriethylenediamine hydroxide [35], and was found to be 903.

### 2.3. Xylanases Pretreatment of Unbleached Rice Straw Pulp

Pretreatment of neutral sulfite unbleached pulp with xylanases was carried out in citrate buffer (pH = 5.3) using 0.04 g of xylanases enzymes per gram of pulp at 10 wt.% consistency for 4 h at 50 °C, as previously described [34]. Chemical composition of the pretreated pulp was 16.46% ash content, 13.18% Klason lignin, 2.31% acid insoluble lignin, 58.4% α-cellulose, 10.79% pentosans, and DP, 1097.

### 2.4. Isolation of Rice Straw Nanofibers (RSNF)

Isolation of RSNF from xylanase-treated unbleached pulp was carried out according to the previously published protocol [34]. In brief, the pulp was first disintegrated using Silverson L4RT shear mixer (Silverson Machines Ltd., Chesham, UK) at 2 wt.% consistency. The pulp was then passed through ultrafine friction grinder (MKCA6-2, Masuko Sangyo, Kawaguchi, Japan) for approximately 140 min. The gap between the disks was adjusted to −90 µm. Chemical composition of the isolated RSNF was 16.8% ash content, 10.51% Klason lignin, 2.11% acid insoluble lignin, 63.5% α-cellulose, 8.84% pentosans, and DP, 1271.

### 2.5. Characterization of RSNF

Transmission electron microscopy (TEM) was carried out using a high-resolution transmission electron microscope (JEM-2100 transmission electron microscope, JEOL, Tokyo, Japan). A drop of fiber suspension was used on a copper grid bearing a carbon film. An acceleration voltage of 100 kV was used. Atomic force microscopy (AFM) was carried out using a VeecoMultiMode scanning probe microscope (Santa Barbara, CA, USA) equipped with a Nanoscope V controller (Veeco instruments, Plainview, NY, USA). A droplet of the aqueous fiber suspension was dried onto a mica surface prior to AFM examination, and images were collected using a tapping mode and a tip model TESPA (antimony (n) doped Si), (Bruker, Camarillo, CA, USA), with a nominal spring constant of 5 N/m and a nominal frequency of 270 kHz. The width was measured on individually separated nanofibers from the height images and the size distribution presented is based on measurements of 50 different nanofibers.

### 2.6. Film Casting

The CA solution (5 wt.%) was prepared by dissolving in acetone. Water in the RSNF suspension (~2 wt.% RSNF) was first removed by vacuum filtration then acetone was passed once through the filtered RSNF. The acetone-wetted RSNF were kept in a closed container in fridge at 8 °C till use. The acetone-wetted RSNF was added to CA solution at ratios from 1.25 to 10 wt.% of dry RSNF to CA; the mixture was homogenized by magnetic stirring for 30 min. The viscosity of the CA/RSNF mixture was measured using a tuning-fork vibration viscometer (Vibro Viscometer SV-10, A&D Company Limited, Tokyo, Japan). The films were prepared by casting/evaporation technique on glass plates and left to dry in air. After drying, the films were annealed at 70 °C for 30 min to remove residual acetone.

### 2.7. Characterization

Mechanical testing in tensile mode was carried out on 1-cm-wide test samples using a Shimadzu universal testing machine (AGX, Shimadzu, Kyoto, Japan) equipped with a 1 kN load cell and a HPV-X2 high-speed video camera to measure strain. Cross-head speed of 2 mm/min was used and the gauge length was 20 mm. Ten samples from each material were measured and the results averaged.

The water contact angle of CA and its nanocomposite films was measured using an EasyDrop measuring system and calculated with the drop shape analysis DSA1 control software, Krüss (EasyDrop Standard, KRÜSS GmbH, Hamburg, Germany), using a sessile drop technique. A 4 μL water drop was placed onto the films at eight separate places for calculating the average contact angle.

Atomic force microscopy (AFM) was used for characterization of topography of the CA films with different RSNF loadings. The measurements were performed on a Veeco Multimode Scanning Probe (Santa Barbara, CA, USA) in tapping mode, with a tip model TESPA (antimony (n) doped Si), Bruker (Camarillo, CA, USA). The root-mean square roughness (RMS) values were measured from the AFM height images and the reported RMS values are the average of five measurements for each sample on a surface area of 225 µm^2^. All measurements were conducted in air at room temperature.

X-ray diffraction (XRD) patterns were recorded on films using an Empyrean X-ray diffractometer (PANalytical, Almelo, The Netherlands).

Microscopic features of films were investigated using a FEI Quanta 200 scanning electron microscope (SEM, FEI Company, Eindhoven, The Netherlands) at an acceleration voltage of 20 kV. Differential scanning calorimetry (DSC) was carried out using Q100 TA (TA Instruments, New Castle, DE, USA). The crystallinity degree of cellulose acetate (X_c_%) was estimated from the ratio between the melting enthalpy of the film under study (ΔH_m_) and the respective value for the totally crystalline material (ΔH^0^_m_), Where ΔH^0^_m_ = 58.8 J/g as follows [36]: Xc = (ΔH_m_/ΔH^0^_m_) × 100.

The porosity (ε) was determined according to the following equation based on the previous gravimetric method that depends on absorption of water by the films [37]:(ε) = [(m1 − m2)/ρ·A·l] × 100
where m1 and m2 are the weight of the wet and dry films, respectively; ρ is the water density (g/cm^3^); A is the effective area of the films (m^2^), and l is the film thickness (m).

Surface area, pore volume, and average pore radius of the films were measured using a Quantachrome Nova-1200 instrument (Quantachrome Instruments, Boynton Beach, FL, USA). The samples were out-gassed overnight at 100 °C prior to measurement.

Dynamic mechanical thermal analysis (DMTA) measurements of the films were carried out using Anton Paar MCR-301 Rheometer (Anton Paar, Graz, Austria) in tensile mode. The measurements were performed at a constant frequency of 1 Hz and strain amplitude of 0.08% in the temperature range of 25 to 250 °C with a heating rate of 3 °C/min and a 20 mm distance between grips.

Thermal stability was studied by thermogarvimetric analysis (TGA) using Perkin Elmer STA 6000 instrument (Perkin Elmer, Waltham, MA, USA).

## 3. Results and Discussion

RSNF isolated from rice straw xylanase-treated unbleached sulfite pulp had very uniform diameter reached that of elementary fibrils [34]. TEM images of the isolated RSNF showed a diameter of about 4 nm, while AMF image showed a diameter of 14 ± 7 nm (Figure 1).

### 3.1. Viscosity of the CA/RSNF Suspension

The viscosity of the CA/RSNF mixture as a function of RSNF content was followed as an indication of the dispersion of RSNF. As shown in Figure 2, the addition of RSNF to the CA solution resulted in a significant increase in its viscosity, indicating good dispersion of the unbleached RSNF. The increase in viscosity ranged from 41% to 146% upon addition of 1.25% to 10% RSNF, respectively.

### 3.2. Mirco-Structure of CA and CA/RSNF Films

Regarding the effect of RSNF on the microscopic structure of CA films, Figure 3 shows SEM images for cross-sections of neat CA film and films containing 1.25–10% RSNF. As shown in the images, the presence of RSNF resulted in drastic changes in the microstructure of the film at loadings above 5%; the cross section appeared as a layered structure while that of neat CA films or CA/RSNF films with a lower nanofiber loading had a compact cross section. This indicates a strong tendency of the nanofibers to agglomerate at high loadings, causing formation of the layered structure.

### 3.3. Transparency and Light Transmittance

Figure 4 shows the visual transparency of the neat CA film and that of a film containing 10 wt.% RSNF on printed paper sheet. The film containing RSNF appeared slightly brownish in color. The thickness of neat CA film was about 21 µm while that of different CA/RSNF samples ranged from 23 to 25 µm. The visual transparency of the CA/RSNF films, even at high contents of RSNF without using compatibilizers or chemical modification, indicates good compatibility between RSNF and CA which could be attributed to the presence of lignin at the surface of the RSNF; lignin has a much more hydrophobic character than cellulose.

Previous work on using bleached CNF with CA showed that white films, rather than transparent ones, were obtained upon using the same casting/solvent evaporation method [33]. This indicates much better dispersion of the unbleached RSNF in the CA matrix used in the current work than in case of using bleached CNF.

Regarding light transmittance, nanocomposite films showed good light transmittance at low RSNF content (<5%), as evident from the measurement by UV-visible spectroscopy (Figure 5); transmittance values of 92%, 87%, and 80% were recorded for neat CA, CA/1.25% RSNF, and CA/2.5% RSNF films, respectively. As the content of RSNF in the films increased, transmittance of the light across the films decreased because of light scattering and absorbance of light due to presence of lignin in the RSNF. In addition, changes in the microstructure of the CA cross section as a result of RSNF addition (Figure 3) could also cause a decrease in light transmittance due to the layered structure formed and the presence of air gaps. Light transmittance values of 50%, 41%, and 27% were recorded for CA/5% RSNF, CA/7.5% RSNF, and CA/10% RSNF films, respectively.

### 3.4. Surface Characteristics: Wettability and Topography

The effect of RSNF addition in various concentrations on CA surface characteristics were assessed in terms of hydrophilicity from water contact angle measurements, and the topographic features in the form of roughness measurements. The roughness and contact angle measurements are presented in Table 1; the values given in the table are the average of both films’ sides. The topographic features obtained from AFM are shown in Figure 6.

The introduction of RSNF resulted in a decrease in the contact angle of CA, as observed from Table 1. Furthermore, the hydrophilicity of the CA films was overall gradually enhanced with increased loadings of RSNF as observed from the decreased contact angles in Table 1. This was expected, as the RSNF network has been shown in a previous study to be hydrophilic in nature, even with such a high lignin content [34]. The contact angle of neutral sulfite xylanase-treated nanofiber networks was 63.9 ± 1.8 [34].

From Table 1, it was also observed that the RSNF also introduces roughness immediately upon addition at low loadings, and that the surface roughness was further increased as more RSNF was added. This behavior was also seen in Figure 6, were the topography of the films’ surfaces changed towards rougher structures as RSNF loading increased.

The porosity of the films with various RSNF content is presented in Table 2. The porosity was not enhanced with the addition of RSNF, and appeared unaffected at the increased loadings (Table 2). With comparable porosities, it can be assumed that the surface structure is in contact with the water upon measurement of the contact angle, rather than in contact with a surface structure of small pores filled with air, and thus the Wenzel regime roughness may be valid [38]. The roughness contribution for that regime, assuming that air-filled pores are not dominating the structure, enhances hydrophilicity in the case of hydrophilic surfaces. Thus, the rougher surface of films with increased RSNF loadings is likely contributing to amplify hydrophilicity.

### 3.5. Water Vapor Permeability (WVP)

The effect of RSNF on water vapor permeability (WVP) of CA films was studied and the results are presented in Figure 7. The WVP of CA films was strongly affected by addition of RSNF even at the lowest loadings. WVP increased with the addition of up to 5% of RSNF then tended to decrease at higher loadings; at 5% RSNF addition, the increase in WVP was about 87% as compared to the neat CA film, while at 7.5% and 10% RSNF loading the increase in WVP was 22% and 17%, respectively. The increase in WVP could be understood from the increased hydrophilicity of the films upon adding RSNF as seen from contact angle measurements, while the tendency to decrease at high RSNF loading could be attributed to RSNF agglomeration and formation of blocked paths to water vapor molecules to pass through the film cross section.

### 3.6. Porosity and Surface Area Characteristics of CA/RSNF Films

The effect of RSNF on porosity of CA films was studied by measuring uptake of water by the films [37]. In spite of the change in the cross section micro-structure shown in the SEM images mentioned above, porosity measurement didn’t show significant changes as a result of RSNF addition to CA (Table 2). Since formation of pores is directly related to the rate of solvent evaporation, the obtained results mean that there was no effect of the RSNF on the rate of evaporation of acetone solvent [39]. Surface area and pore volume measurement confirmed the porosity results as no difference between neat CA and the different CA/RSNF samples was found.

### 3.7. Mechanical Properties of CA/RSNF Films

The effect of RSNF on mechanical properties CA films was studied and the results are presented in Figure 8. As shown in the figure, the addition of RSNF did not have a noticeable effect on the mechanical properties of CA. Regarding tensile strength, the addition of RSNF did not show any significant increase until 10%. The Young’s modulus and strain at maximum load showed larger variations at higher RSNF loadings. Also, no significant increase in Young’s modulus was found. Although a trend for increasing the strain at maximum load at high RSNF loading was found but the variation in the values was high. The absence of the expected high reinforcing effect with increasing the RSNF could be due to the change in the microstructure of the films, as seen in Figure 3, where the cross section of the films became much less compact and with a layered structure at RSNF loadings of more than 5%.

### 3.8. Crystallinity

The crystallinity of the prepared CA/RSNF films was briefly studied using XRD diffraction. CA is a semi-crystalline polymer. As shown in Figure 9, the XRD pattern of CA (degree of substitution, DS ~2.5) showed peaks at two regions; the first peak at about 2θ = 20°, known as the van der Waals or amorphous region, and the second at about 2θ = 8°–10°, which is known as the low van der Waals halo and attributed to the existence of regions of aggregates of parallel chain segments and characteristic of the semi-crystallinity of CA [40]. Addition of RSNF to CA did not affect its crystalline pattern, and at high ratios of RSNF, peaks belonging to cellulose at 2-theta of 18° and 22.5° started to appear. Calculating the crystallinity of CA/RSNF films was not possible because of the overlapping of the peaks of CA with those of RSNF. Crystallinity was estimated from DSC curves as discussed below.

### 3.9. Thermal Properties of CA/RSNF Films

Thermal properties of CA/RSNF films were studied using thermogravimetric analysis (TGA), differential scanning calorimetery (DSC), and dynamic mechanical thermal analysis (DMTA). At first, thermal stability was checked to assure non-degradability of the prepared films in the temperature ranges used in DSC and DMTA analyses. Figure 10 shows thermogravimetry (TG) and differential thermogravimetry (DTG) curves of CA and CA/RSNF with different nanofibers contents. In spite of its low crystallinity, CA with a high degree of substitution has higher thermal stability than cellulose due to the presence of acetyl groups [41]. Thermal degradation of CA involves degradation of the cellulose backbone by decomposition of glycosidic linkages, depolymerization, dehydration, and the loss of acetate groups [42]. RSNF contains amorphous polymers—e.g., hemicellulose and lignin—in addition to the partially crystalline cellulose polymer, which has higher thermal stability than lignin and hemicelluloses [43]. As shown in the figure, CA shows the onset of degradation at a temperature of about 310 °C, while CA containing 2.5% and 10% RSNF showed an onset of degradation temperatures at about 291 and 285 °C, respectively. The lower onset of degradation of CA/RSNF compared to CA could be due to the lower thermal stability of RSNF components. A similar trend was found in the case of the addition bleached cellulose nanofibers to CA, where increasing the cellulose nanofibers content resulted in decreasing the onset of the degradation temperature of CA [33].

Regarding thermal properties, as shown in Figure 11, the DSC curve of CA showed a broad and weak endothermic peak centered at about 120 °C, which could be attributed to loss of adsorbed water by evaporation; a transition at about 187 °C, which could be attributed to T_g_ of CA; and another endothermic peak at about 231 °C, which could be attributed to melting of CA [40,44,45]. Addition of RSNF to CA resulted in shift of the T_g_ peak to about 191, 191, and 198 °C in case of samples containing 2.5%, 5%, and 10% RSNF, respectively, indicating an interaction between RSNF and CA (enlarged areas of the curves are shown in Appendix A). On the other hand, the endothermic peak due to evaporation of adsorbed water became more intense due to the higher moisture sorption and water holding capacity of RSNF; the intensity of that peak increased with increasing the RSNF loading, i.e., more adsorbed water. The peak due to melting of CA at 231 °C did not show significant change as a result of addition of RSNF. The estimated crystallinity of CA films calculated from the melting enthalpy (ΔH) was 12.3% while that of films containing RSNF ranged from 12.4% to 14.8% indicating no significant effect of RSNF on the crystallinity of CA.

DMTA curves (storage modulus and damping factor) of CA and CA/RSNF films are shown in Figure 12. The storage modulus of CA showed a sharp decrease at about 188 °C due to a major relaxation process (known as α relaxation) which is due to a glass-rubbery transition [33]. The addition of RSNF resulted in an increase in the storage modulus of CA before and after T_g_, indicating a slight reinforcing effect of the nanofibers on the CA matrix; this was more obvious at a RSNF content of >5%. Tan delta curves showed corresponding peaks at the T_g_ transition; the peak was at 202 °C in case of neat CA while it was at 208–210 °C for samples containing 2.5–10% RSNF. It is worth mentioning that, in a previous study on nanocomposites consisting of cellulose nanofibers from bleached pulp with CA, addition of cellulose nanofibers to CA resulted in a lowering of its T_g_ value, and no obvious reinforcement of the CA matrix at or above that T_g_ was observed [33]. This may indicate stronger interfacial interaction between the nanofibers containing lignin used in the current work than that isolated from bleached pulp, i.e., without lignin [33]. It is also noted that the intensity of the Tan delta peaks decreased with increasing the RSNF content, probably because of the decrease in CA content since the damping of CA is higher than that of the nanofibers due to the higher viscous nature of the former.

## 4. Conclusions

RSNF with high lignin content could be used to prepare cellulose acetate nanocomposites films with acceptable transparency and improved hydrophilicity using the casting technique. Viscosity measurements of the CA/RSNF suspension indicated a good distribution of RSNF in CA. Addition of the RSNF did not bring noticeable increases in the mechanical properties of the cellulose acetate films, their porosity, or crystallinity. Yet, the microscopic structure of the films was altered, where a layered internal structure was observed at RSNF loadings above 5%, which negatively affected light transmittance across the CA/RSNF films. The presence of RSNF in CA increased the hydrophilicity, water vapor permeability, and roughness of the films, even at low RSNF loadings. Furthermore, the addition of RSNF slightly decreased the thermal stability of CA and slightly shifted its T_g_ to higher values. The presence of lignin allowed compatibility between RSNF and cellulose acetate without modification of the former or adding compatibilizers, thus demonstrating an environmentally-benign approach to influence the structure and performance of cellulose acetate films.

## Figures and Tables

**Figure 1 materials-12-00595-f001:**
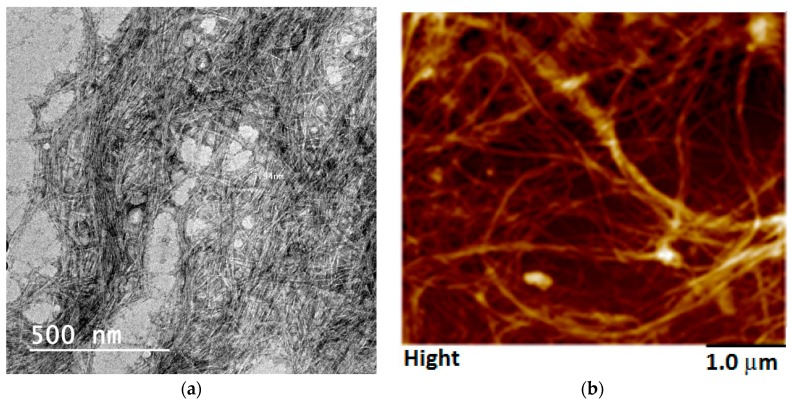
TEM image (**a**) and AFM image (**b**) of RSNF isolated from xylanase-treated rice straw unbleached neutral sulfite pulps.

**Figure 2 materials-12-00595-f002:**
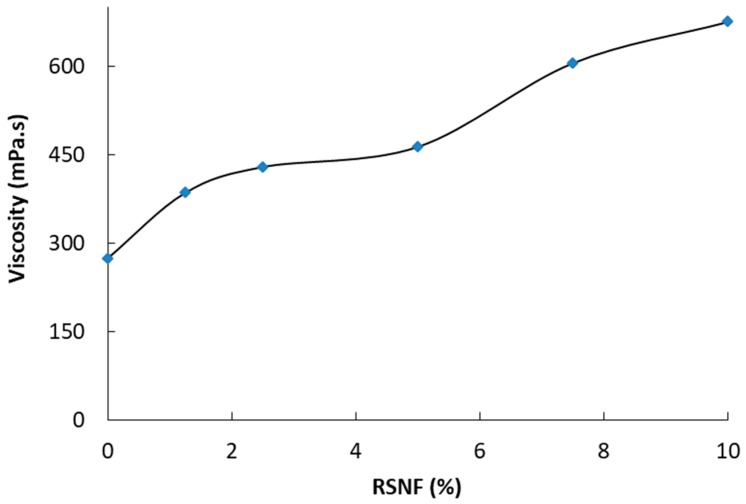
Viscosity of CA solution as a function of RSNF content.

**Figure 3 materials-12-00595-f003:**
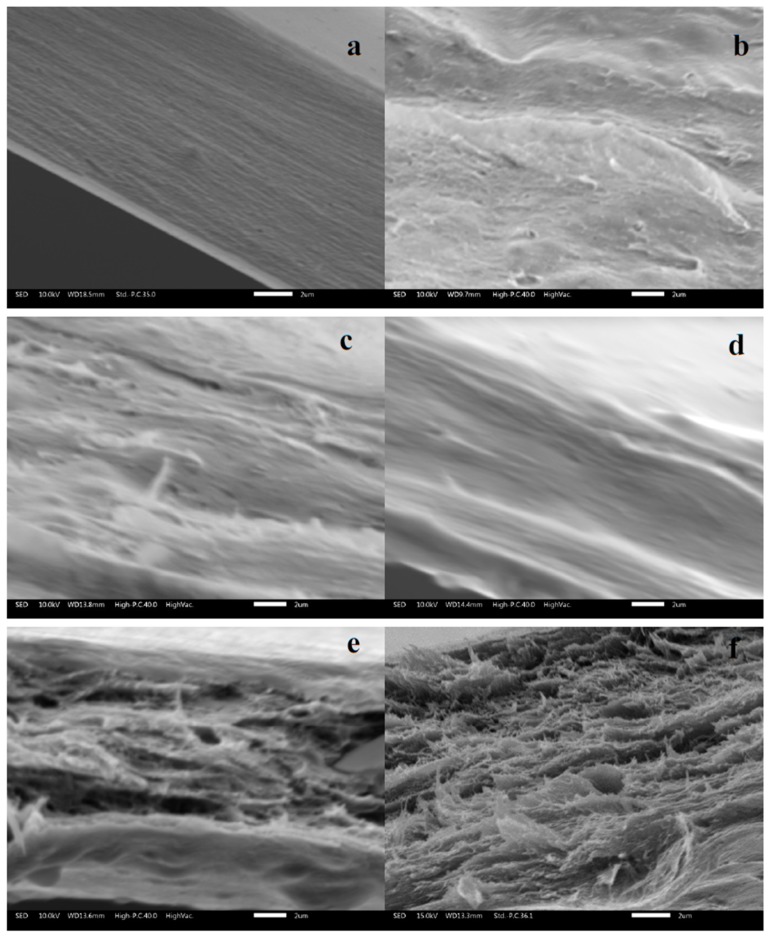
SEM images of (**a**) CA film, (**b**) CA/1.25% RSNF, (**c**) CA/2.5% RSNF (**d**) CA/5% RSNF, (**e**) CA/7.5% RSNF, and (**f**) CA/10% RSNF. The scale bar is 2 µm.

**Figure 4 materials-12-00595-f004:**
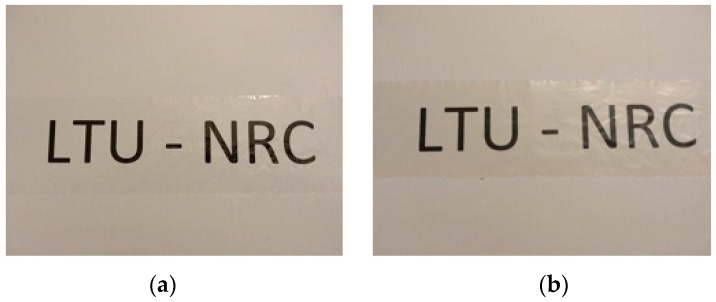
Photos of neat CA (**a**) and CA/10% RSNF films (**b**) over a printed paper.

**Figure 5 materials-12-00595-f005:**
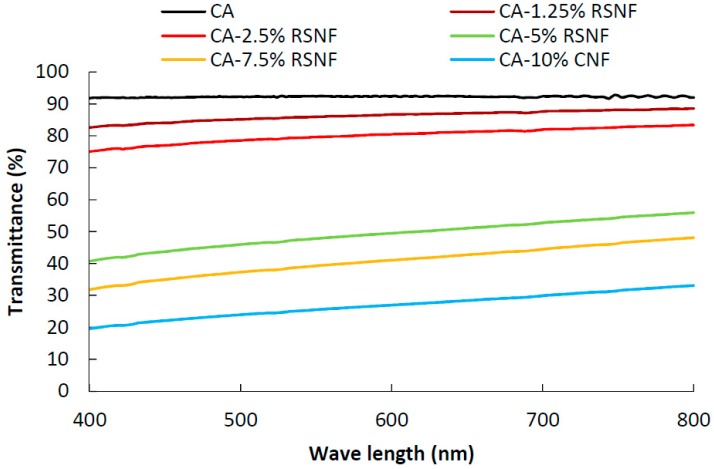
Visible light transmittance of CA films containing different ratios of RSNF.

**Figure 6 materials-12-00595-f006:**
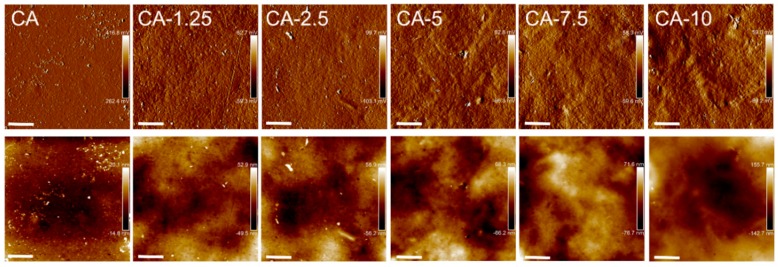
AFM amplitude (above) for CA with different RSNF content and their corresponding height images (below) from where roughness was measured. Scale bar is 3 µm.

**Figure 7 materials-12-00595-f007:**
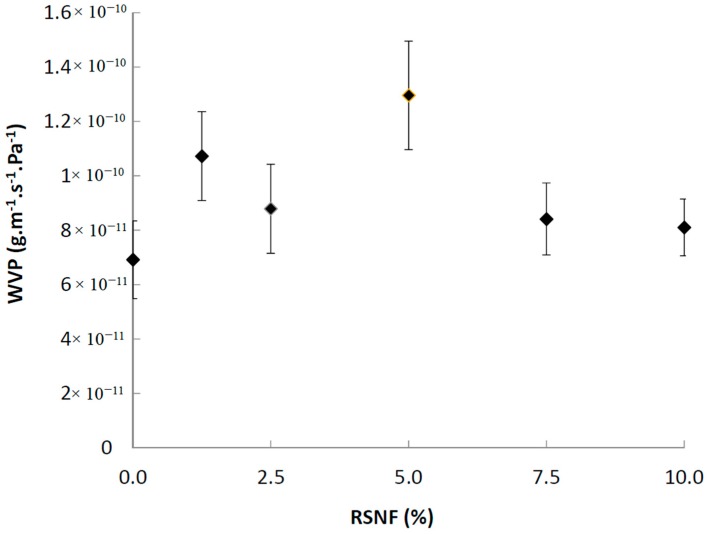
Water vapor permeability (WVP) of CA films containing different ratios of RSNF.

**Figure 8 materials-12-00595-f008:**
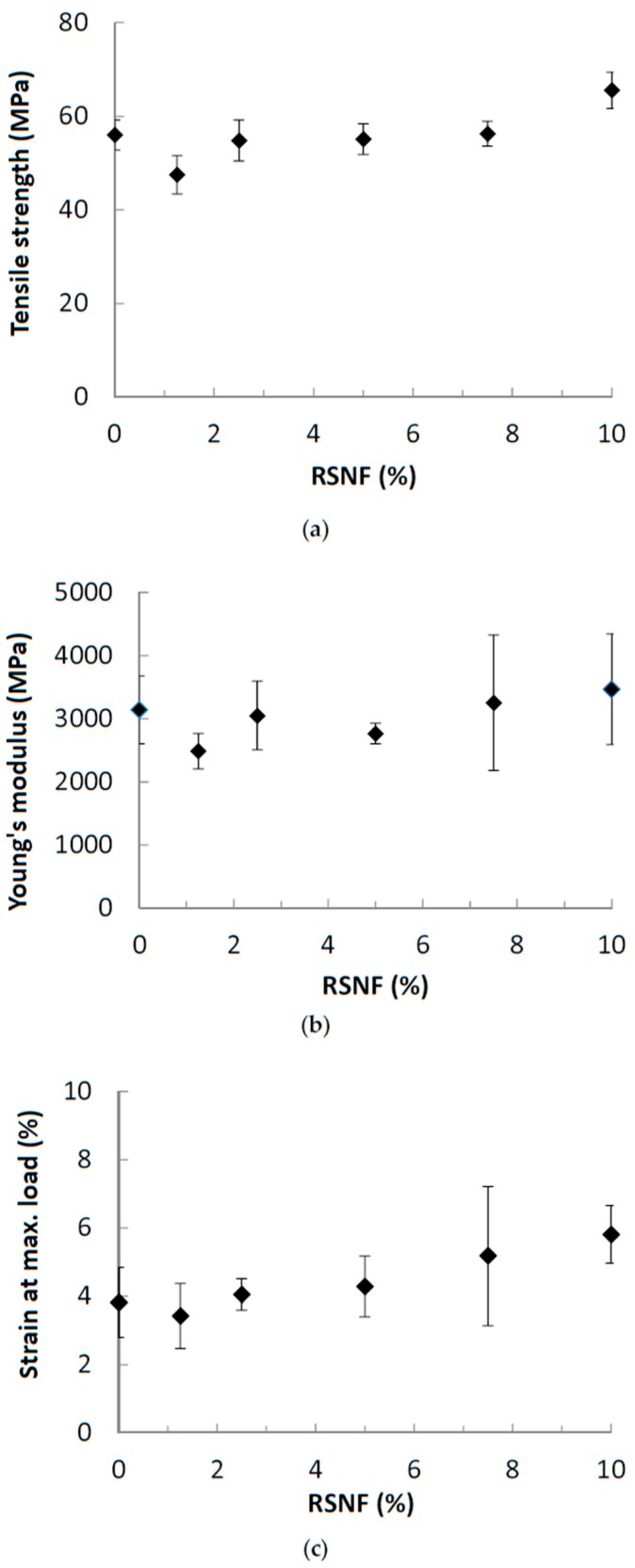
Tensile strength properties of CA films containing different ratios of RSNF contents. (**a**) tensile strength, (**b**) E-modulus, and (**c**) strain to failure.

**Figure 9 materials-12-00595-f009:**
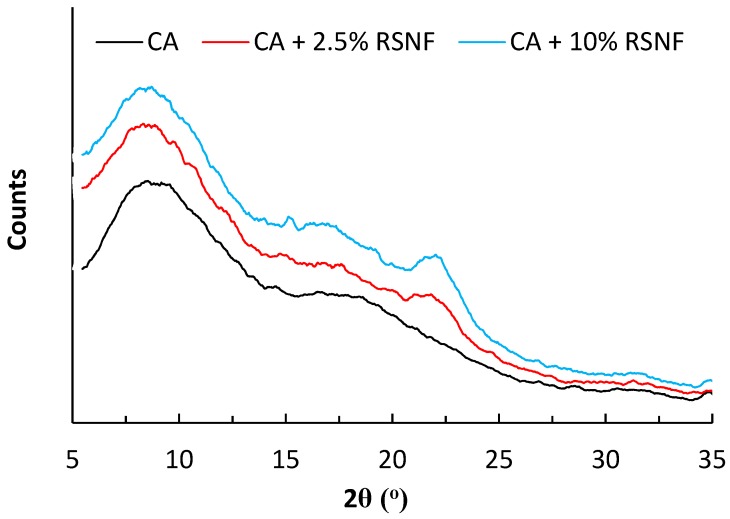
XRD patterns of CA films containing different ratios of RSNF.

**Figure 10 materials-12-00595-f010:**
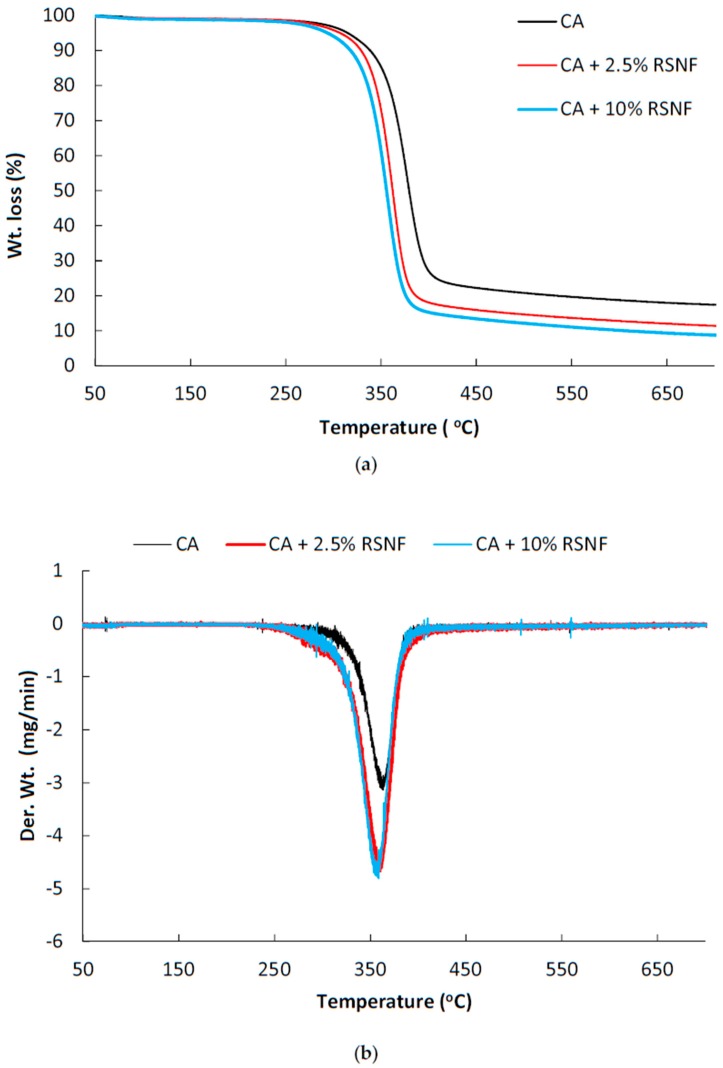
TGA and differential thermogravimetric analysis (DTGA) curves of (**a**) CA and (**b**) CA/RSNF films.

**Figure 11 materials-12-00595-f011:**
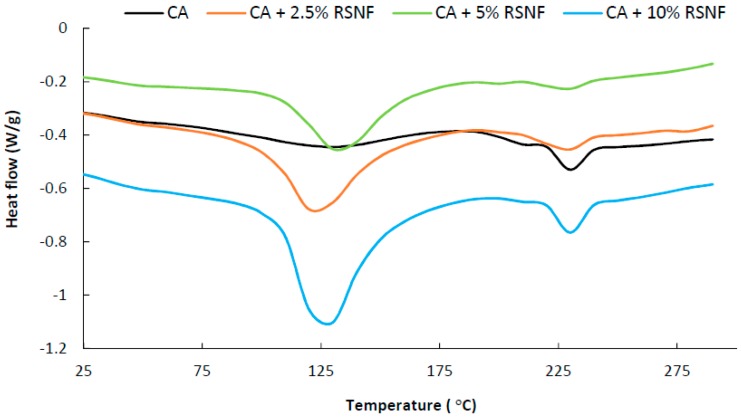
DSC curves of CA and CA/RSNF films.

**Figure 12 materials-12-00595-f012:**
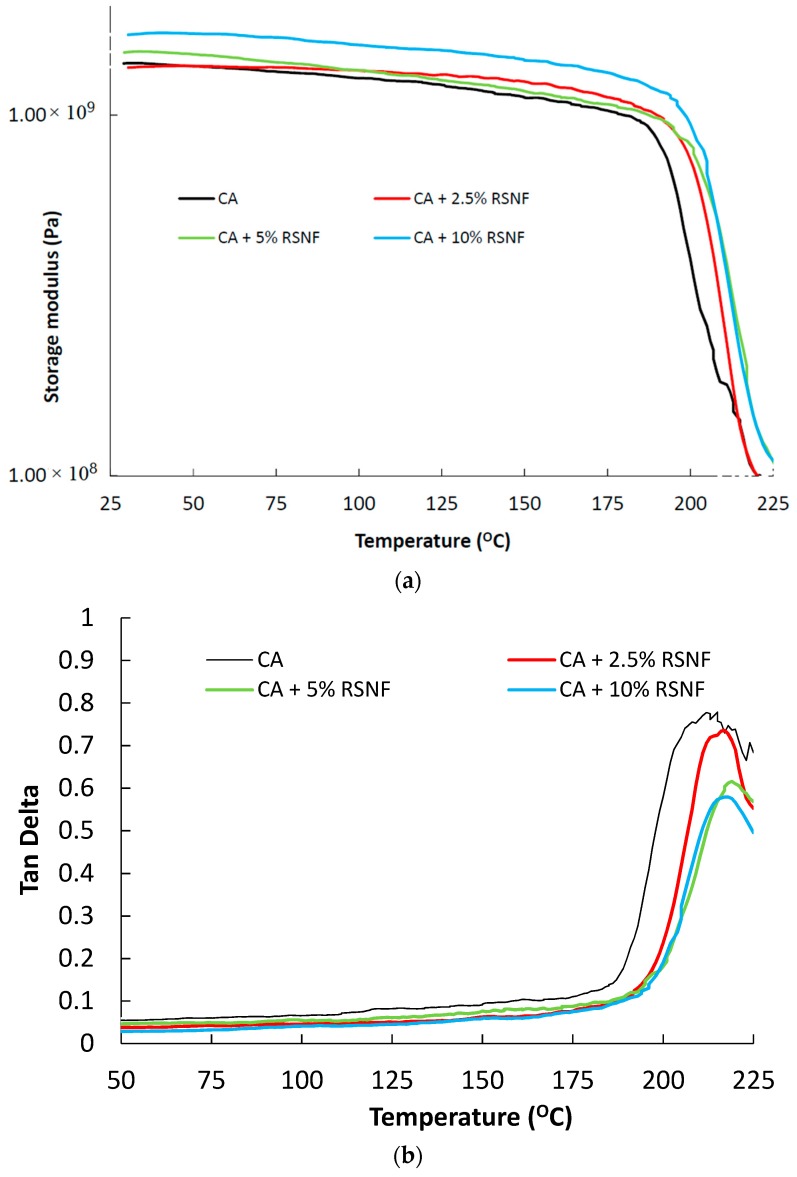
DMTA curves of CA and CA/RSNF films.

**Table 1 materials-12-00595-t001:** Contact angle and roughness measurements of CA and CA/RSNF films.

RSNF Content (wt.%)	CA	CA-1.25	CA-2.5	CA-5	CA-7.5	CA-10
Contact Angle (°)	71.0 ± 2.0	66.1 ± 0.8	66.4 ± 1	65.0 ± 2.2	63.1 ± 2.0	60.7 ± 2.2
Roughness (nm)	5.9 ± 1.9	14.2 ± 2.0	18.0 ± 2.2	20.1 ± 4.0	28.7 ± 5.1	45.9 ± 6.0

**Table 2 materials-12-00595-t002:** Porosity and surface area characteristics of CA and CA/RSNF films.

Property	CA	CA-1.25% RSNF	CA-2.5% RSNF	CA-5% RSNF	CA-7.5% RSNF	CA-10% RSNF
Porosity (%)	15.1 ± 0.72	14.8 ± 0.55	14.6 ± 1.38	14.3 ± 1.1	16.6 ± 0.58	15.1 ± 0.87
Surface Area (m^2^/g)	7.68	-	7.46	6.43	-	8.24
Pore Volume (cc/g)	0.01	-	0.01	0.01	-	0.01
Average Pore Radius (nm)	1.92	-	1.93	1.93	-	1.92

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
