# Peer review of "Nanocomposite Film Based on Cellulose Acetate and Lignin-Rich Rice Straw Nanofibers"

_materials, 2019, doi:10.3390/ma12040595_

Round 1
Reviewer 1 Report
This manuscript focused on the cellulose acetate (CA)-based nanocomposite film containing lignin-rich nanocellulose obtained from rice straw. The presence of lignin allowed compatibility between nanocellulose and CA without any modification, which resulting in successful preparation of nanocomposites via environmentally-friendly approach. Various fundamental data of the obtained composite was shown in this manuscript, however, I cannot understand the scientific importance and impact, or advantage point on material development. In addition, there are many mistakes regarding figure number and reference number in the main text. The author should check them. Thus, I cannot recommend this article for publication at the present stage.
Author Response
Reviewer #1
This manuscript focused on the cellulose acetate (CA)-based nanocomposite film containing lignin-rich nanocellulose obtained from rice straw. The presence of lignin allowed compatibility between nanocellulose and CA without any modification, which resulting in successful preparation of nanocomposites via environmentally-friendly approach. Various fundamental data of the obtained composite was shown in this manuscript, however, I cannot understand the scientific importance and impact, or advantage point on material development. In addition, there are many mistakes regarding figure number and reference number in the main text. The author should check them. Thus, I cannot recommend this article for publication at the present stage.
Response:
- Regarding the scientific importance and impact, thanks a lot for that important comment. We clarified more that in the revised manuscript as follows: Investigating properties of nanocomposites produced using nanofibers with high lignin content is thus important to see what are advantages and disadvantages of these nanofibers. For the best of our knowledge, use of cellulose nanofibers with high lignin content to make cellulose acetate nanocomposites films by casting without chemical modification of the nanofibers has not been studied so far. Using cellulose nanofibers isolated from unbleached pulp in composites reduces the cost when compared to those isolated from bleached pulps. In addition, presence of lignin covering the surface of cellulose nanofibers makes their surface properties different from those isolated from bleached pulps, and thus affect the properties of nanocomposites. The presence of lignin at the surface of the nanofibers can also improve dispersion of cellulose nanofibers in polar aprotic solvents and also improve compatibility with the relatively less hydrophilic polymers like CA. In addition, the presence of lignin with nanofibers allows easier removal of water and its exchange with polar aprotic solvents using simple washing rather than the exhaustive solvent exchange needed in case of the nanofibers obtained from bleached pulps.
- Regarding the mistakes in figures numbers and references, they have been corrected and the whole manuscript has been revised.

Reviewer 2 Report
How the modification of cellulose acetate sheet membrane affects its physical properties as transparency, wettability, topography, water vapor permeability, porosity, mechanical and thermal properties, were measured and discussed. This paper is clearly formulated, its message, its structure are, generally acceptable. However it is not given what are the real aims of these experimental works. Basically, the properties of the nanocomposite film layer did not changed essentially as a function of the nanofiber concentration. A few percentages of changes in the film do not mean improvements/deteriorations of its properties. The composition of the xylanase pretreated nanofibers (its cellulose, hemicellulose and lignin content) is also recommended to be given. Other remarks to be improved:
1. There are lots of typing mistake: see e.g. lines 132, 199, 328, 339, etc.
2. Numbering of figures are not correct after Figure 7: thus e.g. Figure 6 in line 276 is correctly: Figure 8;
3. It is not defined what does mean abbreviation RSNF; I could not find out it; when you use RSNF for the nanofiber used for modification of the cellulose acetate film, then please used it consistently: e.g. see line 302: …cellulose nanofiber films”
4. What does mean: DS (line 292)?
5. Line 206: “…with hydrophobic cellulose acetate”. In reality the cellulose acetate is rather a hydrophilic material; please refine this statement in the whole text;
6. Line 329: where is the appendix 1?
7. Line 327: please mark the Tg peaks in Figure 9 (correctly 11); they cannot be seen in these curves;
8. Conclusion: it is recommended to formulate it by more concrete results;
There are lots of measured data in this paper, which seem to be correct. Thus, it can be recommended, after its improvements, for publication, but the formulation of the real goals of this investigation and also what were, from those, achieved by these results should be clearly presented to the readers.
Author Response
Reviewer # 2.
This paper is clearly formulated, its message, its structure are, generally acceptable. However it is not given what are the real aims of these experimental works. Basically, the properties of the nanocomposite film layer did not changed essentially as a function of the nanofiber concentration. A few percentages of changes in the film do not mean improvements/deteriorations of its properties. The composition of the xylanase pretreated nanofibers (its cellulose, hemicellulose and lignin content) is also recommended to be given. Other remarks to be improved:
Response: Thanks for the valuable and important comments. Regarding the real aims and motivation please refer to the answer written above for the first reviewer who also had similar comment. The composition of the xylanase pretreated nanofibers has been also added.
1. There are lots of typing mistake: see e.g. lines 132, 199, 328, 339, etc.
Response: Sorry for these mistakes, they have been corrected and the whole manuscript has been revised.
2. Numbering of figures are not correct after Figure 7: thus e.g. Figure 6 in line 276 is correctly: Figure 8;
Response: Sorry for these mistakes, numbers of the figures has been corrected.
3. It is not defined what does mean abbreviation RSNF; I could not find out it; when you use RSNF for the nanofiber used for modification of the cellulose acetate film, then please used it consistently: e.g. see line 302: …cellulose nanofiber films”.
Response: Definition of RSNF (rice straw nanofibers) is now placed when first appeared in the abstract. The whole manuscript has been revised and the term RSNF is now unified in the whole manuscript.
4. What does mean: DS (line 292)?
Response: DS is degree of substitution, it is already written before the abbreviation.
5. Line 206: “…with hydrophobic cellulose acetate”. In reality the cellulose acetate is rather a hydrophilic material; please refine this statement in the whole text;
Response: Thanks for the suggestion; we agree that CA is much less hydrophilic in nature than cellulose nanofibers. The term hydrophobic has been changed to relatively less hydrophobic than cellulose and the whole manuscript has been revised.
6. Line 329: where is the appendix 1?
Response: Appendix 1 has been uploaded.
7. Line 327: please mark the Tg peaks in Figure 9 (correctly 11); they cannot be seen in these curves.
Response: Appendix 1 contains the enlarged parts of Figure 11 and shows more clearly how the Tg peaks was determined.
8. Conclusion: it is recommended to formulate it by more concrete results;
Response: The conclusions have been revised.

Reviewer 3 Report
In this contribution, authors report on nanocomposite film based on cellulose acetate and lignin-rich rice straw nanofibers. Materials were deeply characterized by AFM, XRD, TGA/DSC and their physico-mechanical properties fully analyzed.
This a very interesting topic that combines materials and bio-based polymers (cellulose acetate and lignin). Overall, results are highly relevant to the field and the manuscript is nicely written and detailed, recommended almost as such for publication in Materials Journal, only some polishing for typos/mistakes in text and references and maybe Figures to be prepared in better resolution (especially Figures 5, 10, 11 and 12). Otherwise, a nice work recommended for publication.
Author Response
Reviewer # 3
In this contribution, authors report on nanocomposite film based on cellulose acetate and lignin-rich rice straw nanofibers. Materials were deeply characterized by AFM, XRD, TGA/DSC and their physico-mechanical properties fully analyzed.
This a very interesting topic that combines materials and bio-based polymers (cellulose acetate and lignin). Overall, results are highly relevant to the field and the manuscript is nicely written and detailed, recommended almost as such for publication in Materials Journal, only some polishing for typos/mistakes in text and references and maybe Figures to be prepared in better resolution (especially Figures 5, 10, 11 and 12). Otherwise, a nice work recommended for publication.
Thanks a lot for your positive recommendation and the effort done in reviewing the manuscript. Response: the whole manuscript has been revised and the mentioned figures have been replaced by better resolution ones.

Round 2
Reviewer 1 Report
Authors’ response to reviewer and revision of the manuscript is convincing. But, several revisions are still needed before publication.
-Author mentioned that RSNF had uniform diameter and the diameter is ca. 4nm. But, TEM images (Fig. 1) shows huge bundles and aggregates of nanofibrils as main objects. Such bundles were made in drying process for TEM observation? If so, sample concentration of droplet on grid was too high.
-To my knowledge, surface characteristics of cast films are influenced by a substrate. So, author should show information of substrate used for film casting. Which surface of film (substrate side or air side) is analysed in CA measurements and AFM observation?
-Fig. 4 shows high transparency of CA/10% RSNF film. Is this consistent with data in Fig. 5 which shows 20-30% transmittance?
Author Response
Authors’ response to reviewer and revision of the manuscript is convincing. But, several revisions are still needed before publication.
-Author mentioned that RSNF had uniform diameter and the diameter is ca. 4nm. But, TEM images (Fig. 1) shows huge bundles and aggregates of nanofibrils as main objects. Such bundles were made in drying process for TEM observation? If so, sample concentration of droplet on grid was too high.
Response:
Thanks a lot for this comment. This is common in TEM of cellulose nanofibers. The TEM image has been replaced with a better one which shows better individual nanofibers in spite of the aggregation due to drying. An AFM image which shows the nanofibers more clearly has been added. The details of the AMF measurement have been added in the experimental part.
-To my knowledge, surface characteristics of cast films are influenced by a substrate. So, author should show information of substrate used for film casting. Which surface of film (substrate side or air side) is analysed in CA measurements and AFM observation?
Response:
Thanks for your comment. Films were casted on glass plate. The values given for measurements of contact angle and AFM observation are the average of both films’ sides. All these have been clarified in the experimental and results.
-Fig. 4 shows high transparency of CA/10% RSNF film. Is this consistent with data in Fig. 5 which shows 20-30% transmittance
Response:
Thanks for this comment too. Visual transparency of the films over printed paper sheet by naked eye is somewhat different from light transmittance values. The lower values for light transmittance. This was clarified and interpreted in the discussion (lines 216-224; Regarding light transmittance, nanocomposite film showed good light transmittance at low RSNF content (<< span=""> 5%) as evident from the measurement by UV-visible spectroscopy (Figure 5); transmittance values of 92, 87, and 80% were recorded for neat CA, CA/1.25% RSNF, and CA/2.5% RSNF films, respectively. As the content of RSNF in the films increased, transmittance of the light across the films decreased because of light scattering and absorbance of light due to presence of lignin in the RSNF. In addition, the change in the microstructure of CA cross-section as a result of RSNF addition (Figure 3) could also cause a decrease in light transmittance due to the layered structure formed and presence of air gaps. Light transmittance values of 50, 41, and 27% were recorded for CA/5% RSNF, CA/7.5% RSNF, and CA/10% RSNF films, respectively)
Reviewer 2 Report
Authors has written that the conclusion is also revised. This is not done. Thus I recommend to complete with it.
Author Response
Authors has written that the conclusion is also revised. This is not done. Thus I recommend to complete with it.
Response: Thanks for the comment. Conclusions have been further revised and modified. The changed parts are in red color.